# Perceived socio-economic impacts of the marbled crayfish invasion in Madagascar

**Ranja Andriantsoa[1], Julia P. G. Jones[2], Vlad Achimescu[3], Heriniaina Randrianarison[4], Miary Raselimanana[4], Manjary Andriatsitohaina[4], Jeanne Rasamy[4], Frank Lyko[1]***

1 Division of Epigenetics, DKFZ-ZMBH Alliance German Cancer Research Center (DKFZ), Heidelberg, Germany, 2 School of Natural Science, Bangor University, Bangor, United Kingdom, 3 School of Social Science, Mannheim University, Mannheim, Germany, 4 Mention Zoologie et Biodiversité Animale, Université d'Antananarivo, Antananarivo, Madagascar

* f.lyko@dkfz.de

**Data Availability Statement:** All relevant data are within the manuscript and its Supporting Information files.

## Abstract

The negative environmental and economic impacts of many invasive species are well known. However, given the increased homogenization of global biota, and the difficulty of eradicating species once established, a balanced approach to considering the impacts of invasive species is needed. The marbled crayfish (*Procambarus virginalis*) is a parthenogenetic freshwater crayfish that was first observed in Madagascar around 2005 and has spread rapidly. We present the results of a socio-economic survey (n = 385) in three regions of Madagascar that vary in terms of when the marbled crayfish first arrived. Respondents generally considered marbled crayfish to have a negative impact on rice agriculture and fishing, however the animals were seen as making a positive contribution to household economy and food security. Regression modeling showed that respondents in regions with longer experience of marbled crayfish have more positive perceptions. Unsurprisingly, considering the perception that crayfish negatively impact rice agriculture, those not involved in crayfish harvesting and trading had more negative views towards the crayfish than those involved in crayfish-related activities. Food preference ranking and market surveys revealed the acceptance of marbled crayfish as a cheap source of animal protein; a clear positive in a country with widespread malnutrition. While data on biodiversity impacts of the marbled crayfish invasion in Madagascar are still completely lacking, this study provides insight into the socio-economic impacts of the dramatic spread of this unique invasive species. "*Biby kely tsy fantampiaviana, mahavelona fianakaviana*" (a small animal coming from who knows where which supports the needs of the family). Government worker Analamanga, Madagascar.

## Introduction

Invasive alien species can have substantial negative effects on the environment, the economy and human health [1–3]. Invasive species are major drivers of change in invaded ecosystems causing changes in habitat structure, function, and biodiversity, and can result in the extinction of species through direct predation or competition for food and space [4–6]. Damage caused by invasions and the management costs for their control can be a significant economic burden [7]. Also, invasive species can play important roles in disease transmission and spread [3]. While previous

**Funding:** The author(s) received no specific funding for this work.

**Competing interests:** The authors have declared that no competing interests exist.

research has focused on negative impacts, the potential benefits from biological invasions can be overlooked. This may be because negative impacts are part of the definition of invasive species [8]; however there is increasing awareness of their potential positive contributions [9–11]. Few studies have explored how perceptions of costs and benefits from an invasion may vary among people living with an invasive species and how this may change over time following an invasion.

Invasive crayfish have been recognized as a major threat to freshwater ecosystems. Indeed, their position in the food web (most are omnivorous) and their capacity to transmit diseases can strongly impact ecosystem functions. The rusty crayfish (*Faxionus rusticus*), for example, can significantly reduce resource availability in aquatic communities leading to biodiversity loss, while the signal crayfish (*Pacifastacus leniusculus*) transmits the crayfish plague in Europe, which has resulted in extirpation of native crayfish from many locations [12,13]. Nonetheless, invasive crayfish species can also play a positive socio-economic role. The red swamp crayfish (*Procambarus clarkii)* supports important commercial fisheries in many countries where it has been introduced [14,15], while the red claw crayfish (*Cherax quadricarinatus*) is an important source of income for artisanal fishermen in Jamaica [16]. Other studies have suggested that invasive crayfish in Kenya play a role in control of schistosomiasis through predation on the snail hosts of this human parasite [17].

The marbled crayfish (*Procambarus virginalis*) originated from the German aquarium trade in the early 2000s [18,19]. Marbled crayfish are descendants of the sexually reproducing slough crayfish (*Procambarus fallax*) from Florida [20]. The ability of the marbled crayfish to reproduce without males, as well as its high fecundity and resistance to harsh environmental conditions, predisposes it to be an invasive species [21,22]. Indeed, marbled crayfish have formed stable wild colonies in several European countries [23–27]. In Madagascar, marbled crayfish were first observed by biologists in markets around 2005 but were not identified until 2007 [22,28]. Initially confined to a small region around Antananarivo, their distribution area has now increased about 100-fold, covering eight out of 15 regions studied in 2017 [29]. Conservationists quickly raised concerns [19,22] about potential impacts that marbled crayfish may have on endemic freshwater biodiversity, including the seven species of endemic crayfish in the genus *Astacoides* [30,31]. In addition, there were also concerns that marbled crayfish may have a negative impact on rice agriculture and local freshwater fisheries. However, farmers were observed transporting the species intentionally around the island as early as 2007, suggesting that they were considered a valued food source [22]. While the Ministry of Agriculture, Livestock and Fisheries issued legislation to prohibit the transportation of live marbled crayfish in 2009, the animals have recently been observed being widely sold in markets across Madagascar, without sanctions [32].

Our study aims to update information on the distribution of marbled crayfish in Madagascar and to explore attitudes towards marbled crayfish in three regions that were chosen to reflect a gradient in terms of how recently the marbled crayfish arrived. Specifically, we explore perceptions of the positive or negative impacts of the crayfish on rice agriculture, fishing and their contribution to health, food security and household income. We also seek to understand more about the markets for this species, including the relative acceptance of marbled crayfish as a source of dietary protein. While information on the ecological impacts of the invasion of marbled crayfish in Madagascar are lacking, this paper provides valuable information on their socio-economic impacts.

## Materials and methods

### Ethics statement

Research involving human participants was approved by the Bangor University College of Environmental Sciences and Engineering research ethics committee (Approval Number:

CoESE2019RAJPGJ01). Oral consent was obtained from each participant to voluntarily participate in the survey. Field research was approved by the Ministry of Environment and Sustainable Development of Antananarivo (MEDD), Madagascar (research permits No. 58/19/MEDD/SG/DGF/DSAP/SCB.Re and No. 59/19/MEDD/SG/DGF/DSAP/SCB.Re).

## Update of marbled crayfish distribution

To update information on the distribution of marbled crayfish in Madagascar, we re-visited 18 locations and 14 additional locations (see S1 Data of S1 Table for details) across three regions where a survey in 2017 [29] did not reveal marbled crayfish (Analanjirofo, Menabe, Vatovavy Fitovinany). We sampled the water bodies (rice fields, channels, ponds and marshes) with traditional "tandroho" nets (50cm x 30 cm x 30cm, S1 Data of S1 Fig) from 8:00 to 11:00 and from 15:00 to 18:00 for three days in each location, as described previously [33]. Collections were performed by M.A. Additionally, we recorded the geographic coordinates of new locations where we observed marbled crayfish within the survey regions: Analamanga, Mahatsiatra Ambony and Ihorombe (S1 Data of S2 Table).

## Social survey details

The survey was carried out in March and April 2019 in three regions where marbled crayfish are present: Analamanga, Mahatsiatra Ambony and Ihorombe (see S1 Data of S2 Fig and S3 Table for details). In each region, we first visited the regional authorities (Direction Regionale de l'Environnement, de l'Ecologie et des Forets, DREEF), as well as local authorities in each town or village where we worked (the commune or fokontany representative) to gain permission for the work. The survey was conducted on both weekdays and weekends.

We collected three types of data. Quantitative information was collected using a brief questionnaire that targeted adults in each region. The questionnaire contained closed-ended items including Likert scale responses. We did not have a sampling frame and used purposive sampling. At each location, enumerators approached adults in markets, fields, streets and villages, or working in or near water bodies. The questionnaire took approximately 15 minutes to complete. Key informants (those involved in selling crayfish, crayfish harvesting or having more knowledge on marbled crayfish), were asked some additional open-ended questions. These in-depth surveys lasted 30–45 minutes. The full questionnaire (in English and Malagasy) is available in the Supporting Information. Lastly, we recorded the price of processed marbled crayfish and other common sources of dietary protein (as price per kilogram) at 30 markets (containing more than 200 stalls in total) in Antananarivo.

We introduced ourselves to each respondent and carefully explained the aim of our study and how data would be used. We provided assurance of confidentiality by explaining that we were not taking any individually identifying information or recording voices. Due to the low level of literacy, oral consent was obtained from each respondent to voluntarily participate in the survey. Oral consent was also obtained for all photographs taken. Interviews were carried out by R.A., J.R., H.R. and M.R. with the help of trained enumerators. All interviewers were native speakers of Malagasy, comfortable in the dialects of the regions visited. Before starting an interview, each informant was shown a series of 16 photographs with crustacean and fish species to assess their knowledge of marbled crayfish (S1 Data of S3 Fig). Surveys were delivered using Open Data Kit (ODK)-Collect, an Android open source app for data collection. Data were submitted to ODK Aggregate when an internet signal was available.

We assessed the perception of marbled crayfish impacts by collecting information on the respondents' socio-economic status and their preferences. First, we showed a series of 16 photographs (S1 Data of S4 Fig) that illustrated local sources of dietary protein. We asked the

respondents to rank them in order of preference and recorded their top five preferences. Next, we collected information on their primary and secondary livelihood (agriculture, government position, private monthly waged work, daily waged labor, trader, wild product harvester, other). We also recorded if people owned irrigated rice fields, and if they fished commercially, for subsistence or recreation. We investigated the perception of marbled crayfish impacts on six different items (rice farming, fishing, household economy, food security, animal feed, health) using a 5-point Likert scale illustrated by Emojis (very positive to very negative, coded 1 to 5 respectively). Three of these items (rice farming, fishing, animal feed) were only presented to respondents that were engaged in these activities, and were not used in the construction of the perception of marbled crayfish index.

## Data analysis

Results from questions with Likert scale responses were illustrated as diverging stacked bar charts. We explored the relationships between marbled crayfish impact perception and potential predictors by fitting a multivariate ordinary least squares (OLS) regression. Our model investigated which factors predicted whether the perception of marbled crayfish impact was more positive or negative.

The dependent variable is an index of local perception towards the crayfish built by standardizing the sum score of the four perceived marbled crayfish impact items. Each of the items used in the perception of marbled crayfish index (impact on household economy, food security, health and overall impact) were measured on a 5-point ordinal scale. The index varies from -2.33 (most negative impact) to 2.24 (most positive impact), with a mean of 0 and standard deviation of 1.

Complete data was available for 288 respondents, while 64 respondents had missing values for one or two of the four items. For these cases, values were imputed using the forward imputation method described in [34], based on nonlinear principal component analysis and nearest neighbor matching. We imputed 6 values for overall impact, 19 values for impact on household economy, 12 values for impact on food security and 38 values for impact on health. We excluded 12 cases with missing data on three or four items. We also excluded three cases with an age of under 18 (not in target population) or over 80 (outliers in the regression).

We checked for the reliability of the combined index by calculating the Cronbach Alpha statistic for the four items used to build the dependent variable. Alpha was 0.80, indicating strong internal consistency (S1 Data of S4 Table). Bivariate item correlations (Spearman's rho) were in the range of 0.4–0.6 (S1 Data of S5 Fig), indicating that the variables are positively associated. Correlations between the index value and each item were over 0.7 (S1 Data of S4 Table). Alpha and correlation coefficients were not significantly different on the imputed and non-imputed dataset.

Eight variables were considered as potential predictors: survey location, gender, age, livelihood, relationship to marbled crayfish (harvester, vendor, customer), and the use of marbled crayfish as animal feed. All variables except age were categorical and transformed into dummy variables before running the regression analysis.

To validate our regression model, we assessed the assumptions of normality, homoskedasticity and absence of multicollinearity. We first assessed the normality of residuals with a histogram, a Q-Q plot (S1 Data of S6 Fig), and a Shapiro-Wilk normality test. We also checked for heteroskedasticity using the Breusch-Pagan test and for multicollinearity using the Variance Inflation Factors (VIF). Finally, we evaluated the prediction model accuracy by running a k-cross validation analysis (10-fold).

Data analysis was carried out using R3.4.3. Charts were produced using the *ggplot2* and *corrplot* packages. Reliability of the dependent variable was checked using the alpha function from the *psych* package. Model checks were performed with functions imported from the *car* package. Imputation was performed using the *ForImp* package. Cross-validation was performed using the *caret* package, with a seed set on 523 for the reproducibility of the analysis (S1 Data of S5 Table).

## Results

Compared to a previous survey in 2017 [29] we did not record an expansion of the known range of the marbled crayfish in Madagascar during this survey (March and April 2019). Re-sampling at 18 previously negative sites, and 14 additional locations distributed in the Analanjirofo, Menabe and Vatovavy Fitovinany regions revealed no marbled crayfish (Fig 1A, S1 Data of S1 Table). However, marbled crayfish were discovered at 15 additional sites in the Analamanga, Mahatsiatra Ambony and Ihorombe regions (i.e., within the previously

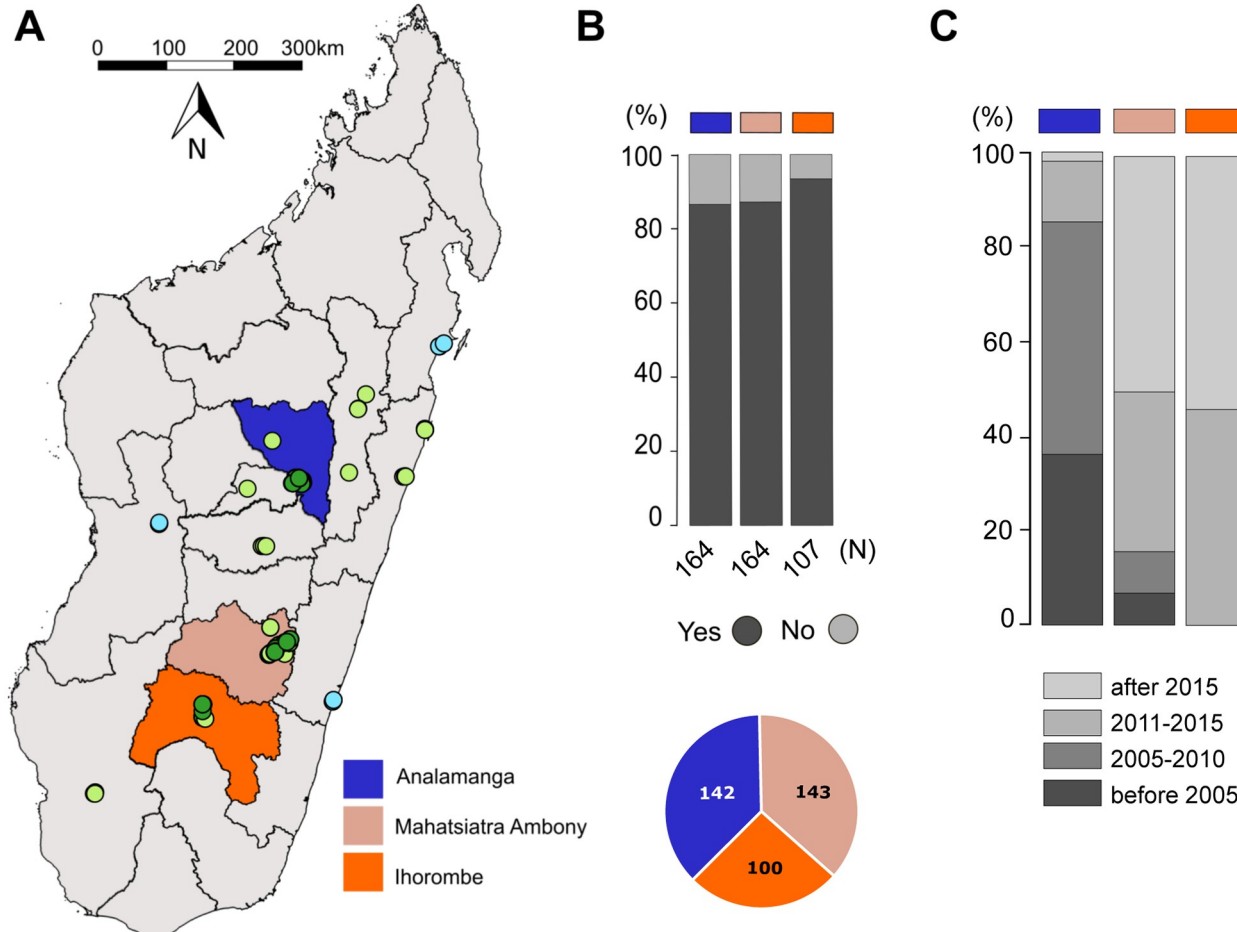

**Fig 1. Distribution of marbled crayfish in Madagascar.** (A) Map of Madagascar. Light green dots indicate sites with known marbled crayfish populations from 2017 [29]. Dark green dots indicate additional sites where marbled crayfish were detected during this study (in 2019), light blue dots indicate 3 re-sampled negative regions from 2017 with no marbled crayfish. The three regions where the social surveys presented in this paper were carried out are highlighted by color. (B) Participant response rates and numbers of survey respondents per region. (C) Earliest observation of marbled crayfish in the three regions. Numbers indicate % respondents.

identified range) during the present survey (Fig 1A, S1 Data of S2 Table), which likely reflects the continuing anthropogenic distribution due to commercial exploitation [32].

We conducted our social research in three regions (Fig 1A, S1 Data of S3 Table): Analamanga (mostly in Antananarivo at 12 sampling locations), Mahatsiatra Ambony (mostly in Fianarantsoa at 8 sampling locations) and Ihorombe (mostly in Ihosy at 3 sampling locations). Using a purposive sampling, we approached a total of 435 adults (Fig 1B) and obtained participation rates of over 86% in each of the three regions (Fig 1B). The sample consisted of 142, 143 and 100 participants in Analamanga, Mahatsiatra Ambony and Ihorombe, respectively. The average participant age was 39 years, with 54% male and 46% female participants. Farming was the most commonly reported primary livelihood among respondents (48%).

A subset of respondents (n = 275) were asked when they first observed marbled crayfish in their area. In the Analamanga region, a substantial fraction of respondents indicated that they became first aware of marbled crayfish in 2005 or before and the animals became well-known by 2010 (Fig 1C). In contrast, marbled crayfish were mostly unknown in Mahatsiatra Ambony and completely unknown in Ihorombe before 2011 (Fig 1C). This is consistent with a spread of the animals from the center to the south of Madagascar.

To explore the perception of marbled crayfish impacts, we used a Likert scale to investigate six variables. Results showed that attitudes towards marbled crayfish varied substantially by region. In general, respondents in the two southern regions (Mahasiatra Ambony and Ihorombe) had more negative attitudes towards the crayfish, while those in Antananarivo were generally more positive (Fig 2). However, in all three regions, respondents reported both positive and negative impacts of marbled crayfish. Negative impacts on rice farming and fishing were reported at all sites (and by nearly 100% of respondents in Ihorombe), while contribution to food security and household economy, in general, were seen as positive.

Responses to the open-ended questions by key informants (a subset of 52 participants) provided further information to support and explain the quantitative data. The reasons for widely reported negative impacts of marbled crayfish on rice fields was due to burrowing activities, which dry up the rice fields and require the farmers to regularly repair their banks and irrigation canals. For example: "the marbled crayfish destroy the mud walls we build around our rice fields" (S1 Data of S6 Table, Q1-3). This can also lead to arguments between rice field owners as they would accuse each other to have dried the rice field by digging burrows.

The occasional positive comments about the impacts of marbled crayfish on rice farming came from the perception that marbled crayfish burrowing activities could improve rice production by loosening and aerating the soil (S1 Data of S6 Table, Q4). Perceived impacts of marbled crayfish on fishing were overwhelmingly negative, because crayfish are thought to predate on young fish (though not larger fish; S1 Data of S6 Table, Q5). Importantly, some impacts may not be directly observed after an initial marbled crayfish colonization: "the impacts on fishing cannot really be seen since the marbled crayfish are present for only two years" (S1 Data of S6 Table, Q6). Many people mentioned positive impacts from crayfish harvesting on local livelihoods from catching and selling crayfish, or using them as a cheap source of household protein or animal feed. A Fokontany worker in a village with several households who depends on marbled crayfish described it as "a small animal coming from who knows where which supports the needs of the family" (S1 Data of S6 Table, Q7). Similarly, a farmer reported that marbled crayfish had given him a good living for about 10 years since they arrived in the area (S1 Data of S6 Table, Q8). Marbled crayfish were often mentioned as a good and cheap source of animal protein, and a source of energy and strength (S1 Data of S6 Table, Q9). However, some respondents have also experienced and witnessed allergic reactions, and stomach pain due to consumption of marbled crayfish (S1 Data of S6 Table, Q10).

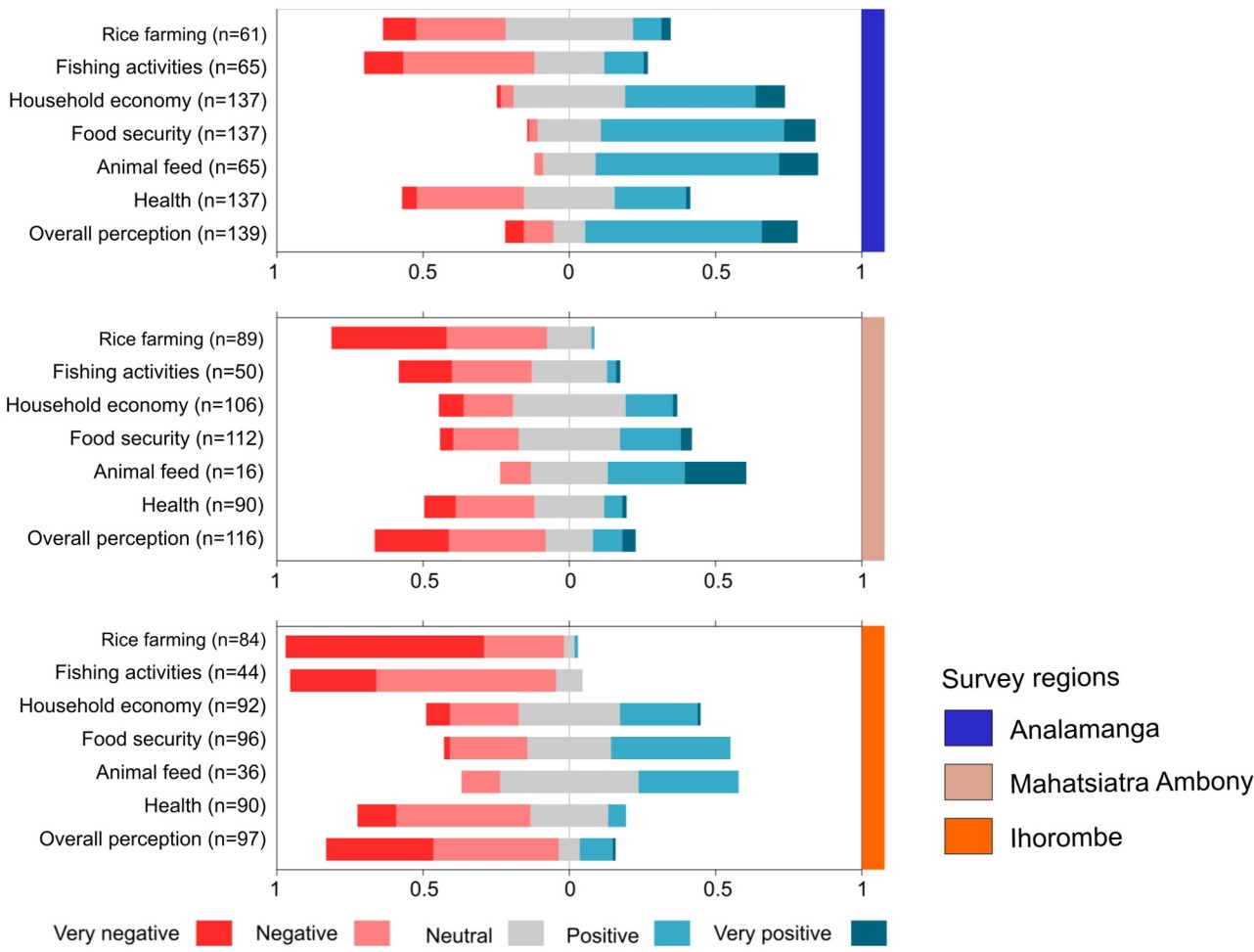

**Fig 2. Perceived socio-economic impacts of marbled crayfish in 3 regions of Madagascar.** Diverging stacked bar plots show overall impacts (Likert scales) in the three regions.

In order to link the perception of marbled crayfish impacts to socio-economic characteristics of respondents, we selected eight variables as predictors in a multivariate linear model. Our linear regression model (OLS) explored the relationship between these variables, and the perception of marbled crayfish index (Fig 3, Table 1). The model shows a good overall fit: it successfully explains 51% of the variation of the dependent variable (adjusted R squared of 0.506). The residuals satisfy the assumption of normality (Shapiro-Wilk normality test on the residuals: p = 0.182), confirmed visually by a Q-Q plot (S1 Data of S6C Fig). There is no observed heteroskedasticity in the data (Breusch-Pagan test chi square statistic = 1.410, p = 0.234). Additionally, 10-fold cross-validation of our regression analysis revealed a good performance with an R-squared of 0.504, providing additional confidence in the results of the regression analysis.

A positive overall perception of marbled crayfish is associated with study site, as participants in Analamanga showed a significant (p<0.001) positive association (+0.749 index score) and Ihosy a marginally significant (p = 0.071) negative association (-0.182 index score) compared to Fianarantsoa (controlling for gender, age and livelihood). Those involved in crayfish harvesting were more positive towards marbled crayfish impacts than non-harvesters (+0.557,

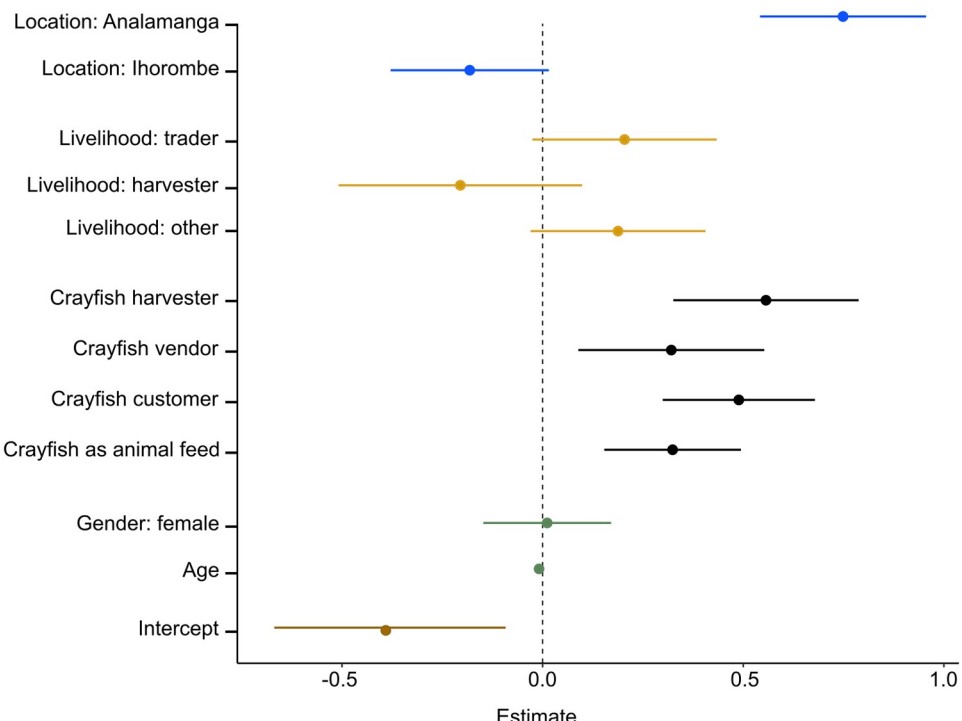

**Fig 3. Coefficient plot of the model exploring the relationship between perception of marbled crayfish index and 8 socio-economic predictors.** Blue bars represent the coefficients for locations while yellow bars show the coefficients for livelihoods, black bars illustrate the coefficients for crayfish-related activities and green bars the coefficient for demographic controls. Error bars represent 95% confidence intervals. Gender: male, and age were included as controls. The intercept was defined by the following parameters: location: Mahatsiatra Ambony; livelihood: farmer, not harvesting crayfish, not selling crayfish, not buying crayfish, not using crayfish as animal feed. P-values of each predictor are provided in Table 1.

p<0.001). A similar scenario was observed for crayfish customers (+0.489, p<0.001), vendors (+0.321, p = 0.007), and those who used marbled crayfish as animal feed (+0.324, p<0.001), compared to those who do not buy, sell or use marbled crayfish respectively. Livelihood was marginally significantly associated with overall perception, with traders (+0.204, p = 0.081), and our grouping of 'other' livelihoods (+0.188, p = 0.091) having a more positive average scores compared to farmers. This finding is not surprising, as farmers will suffer negative impacts incurred by marbled crayfish on rice fields, while traders tend to profit from selling them. Finally, age has a negative association (-0.009 per year, p = 0.008), with older respondents having a less positive view of the overall impact of marbled crayfish.

To assess the popularity of marbled crayfish in the diet of our participants, we asked them to identify and rank their five most preferred sources of nutritional protein among those illustrated in 16 photographs (S1 Data of S4 Fig). Results showed that tilapia and traditionally reared chicken were the most preferred source of dietary protein (Fig 4), which is consistent with previous findings [35,36]. Marbled crayfish meat showed a middling level of preference and was grouped together with shrimp, factory-produced chicken meat and beef liver (Fig 4). Several other dietary protein sources, such as tripe, beans and crabs were less preferred than marbled crayfish.

In Antananarivo (Analamanga region), marbled crayfish were mostly sold in markets (Fig 5A) and their leftover carapaces were used as animal feed (Fig 5B). We found marbled crayfish in 33 stalls in 11 markets (out of 30 markets investigated). They were mainly sold as boiled tail

**Table 1. Regression table: The dependent variable is the perception of marbled crayfish index.**

| Predictors | | Estimate | Std. Error | t-value | p-value |
|---|---|---|---|---|---|
| Location | Antananarivo | 0.749 | 0.105 | 7.114 | 0.000*** |
| | Ihosy | -0.182 | 0.100 | -1.813 | 0.071 |
| Livelihood | Trader | 0.204 | 0.117 | 1.749 | 0.081 |
| | Wild product harvester | -0.205 | 0.154 | -1.329 | 0.185 |
| | Other | 0.188 | 0.111 | 1.696 | 0.091 |
| Crayfish harvester | Yes | 0.557 | 0.117 | 4.740 | 0.000*** |
| Crayfish vendor | Yes | 0.321 | 0.118 | 2.721 | 0.007** |
| Crayfish customer | Yes | 0.489 | 0.096 | 5.072 | 0.000*** |
| Crayfish as animal feed | Yes | 0.324 | 0.087 | 3.740 | 0.000*** |
| Gender | Female | 0.011 | 0.081 | 0.140 | 0.889 |
| Age | Continuous | -0.009 | 0.003 | -3.031 | 0.003** |
| Intercept[2] | | -0.391 | 0.147 | -2.667 | 0.008** |
| Observations | | 354 | | | |
| $R^2$ | | 0.5221 | | | |
| **Adjusted $R^2$** | | **0.5066** | | | |
| Residual Std. Error | | 0.7024 (df = 340) | | | |
| F Statistic | | 33.76 (df = 11; 340) | | | |
| p-value | | < 0.0001 | | | |

Significant at <0.1, *significant at <0.05, **significant at <0.01, ***significant at <0.001.

The perception of marbled crayfish index is built by standardizing the sum score of four Likert scale items (overall impact of marbled crayfish, impact on household economy, impact on food security, impact on health).

Location: Fianarantsoa, Livelihood: Farmer, Not harvesting crayfish, Not selling crayfish, Not buying crayfish, Not using crayfish as animal feed and Gender: Male.

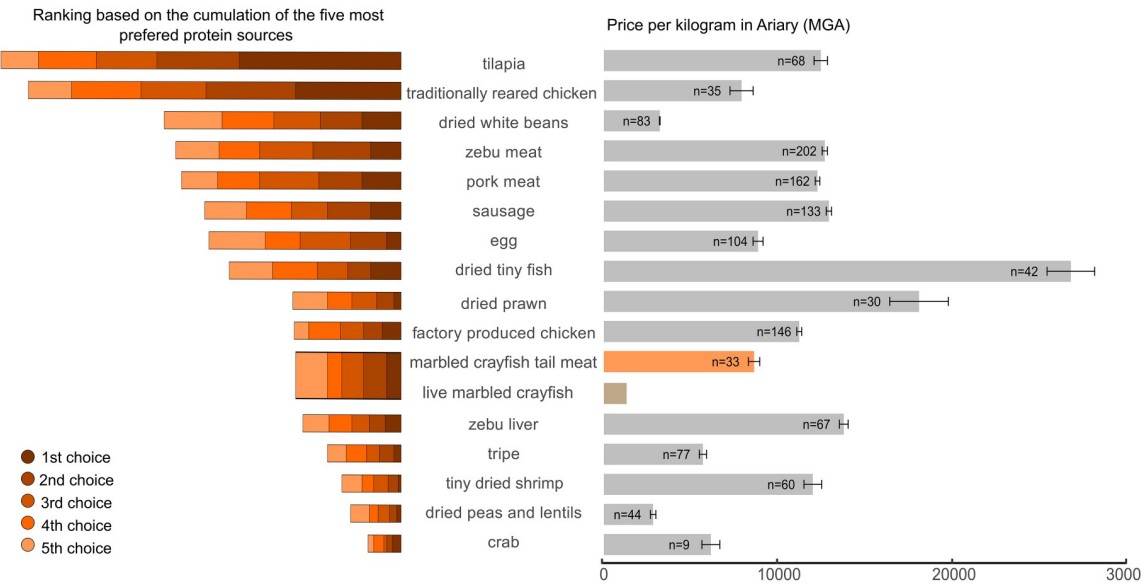

**Fig 4. Preferences and relative prices per kilogram for a selection of common sources of dietary protein.** Photographs and descriptions are provided in S1 Data of S4 Fig. Left panel: results of preference ranking for sources of dietary protein. Right panel: mean of market prices per kilogram for each dietary protein source (n = number of stalls where prices were recorded in the 30 Antananarivo markets visited). Error bars represent 95% confidence intervals. For comparison, the price of live marbled crayfish from vendors in Ihorombe (1,200 MGA/kg, approximately 0.3 EUR) is also indicated.

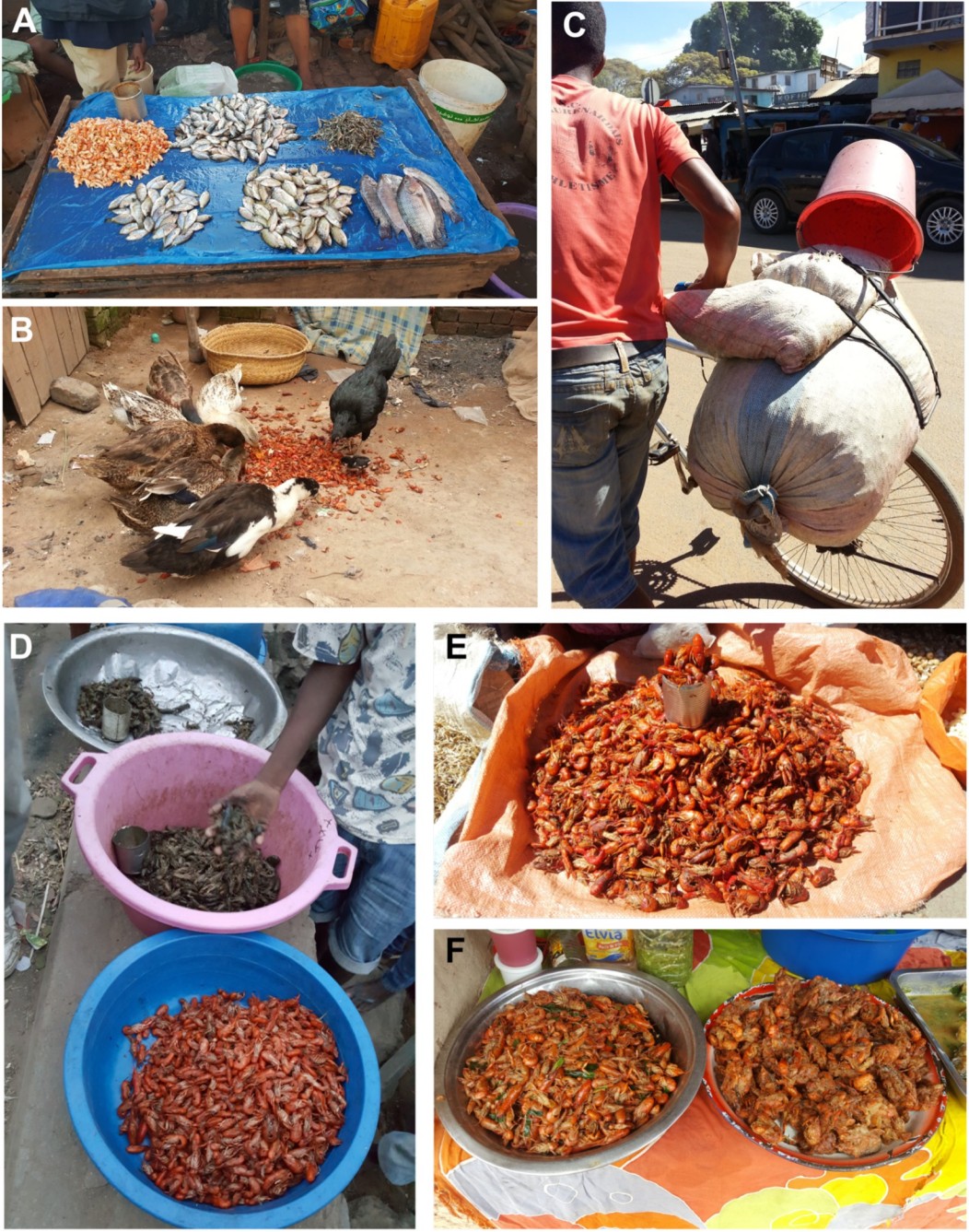

**Fig 5. Commercial sales and distribution of marbled crayfish.** (A) Marbled crayfish meat sold with freshwater fish. (B) Leftover shells from marbled crayfish used as animal feed. (C) Bag of live marbled crayfish for distribution. (D) Live and boiled marbled crayfish sold on the street. (E) Boiled and salted marbled crayfish sold in the market. (F) Ready-to-eat marbled crayfish sold in street food markets.

meat, with an average price per kg of 8,600 MGA (approximately 2 EUR, also see Fig 4). This strongly contrasts with Ihosy (Ihorombe region), where marbled crayfish were predominantly sold live in large bags for widespread distribution (Fig 5C). The marbled crayfish business is structured differently in Ihosy compared to Antananarivo: orders are received from different

locations and regions. Traders then purchase live marbled crayfish from local collectors in response to market demand. Up to 20 bags of 35 to 80 kg with live marbled crayfish are transported daily with short and long-distance "taxi-brousse" for a fixed fee per bag during the rainy season (November to March). The price of live marbled crayfish is much lower with an average of 1,200 MGA/kg (approximately 0.3 EUR).

Interestingly, Fianarantsoa (Mahatsiatra Ambony region) was mentioned as one of the cities where people bought marbled crayfish from Ihosy. This was confirmed by interviews of marbled crayfish vendors in Fianarantsoa's evening market who indicated that they buy marbled crayfish from Ihosy, and usually sold their entire amount within a few hours (Fig 5D). In Fianarantsoa, marbled crayfish were usually sold as processed food (Fig 5E). Marbled crayfish beignets and meals with marbled crayfish in tomato and green onion sauce (Fig 5F) were offered in several street food locations and street market restaurants. Despite an official prohibition from the Ministry of Agriculture, Livestock and Fisheries (effective since 2009) that forbids the transport of live marbled crayfish, the transport of marbled crayfish (for example by taxi-brousse), is supported by regular permits and fees. We interviewed taxi-brousse drivers from different transport companies in Ihorombe, who stated that those who transport marbled crayfish pay for a yearly permit, similar to transport permits for other freshwater products.

## Discussion

It is now indisputable that we live in the Anthropocene; a geological age defined by humanity's impact on the environment [37]. In this situation, some ecologists are rethinking their response to invasive species and increasingly recognize that non-native species are a part of the ecology of the future [38,39]. While the strongly negative impacts of some invasive species render this a controversial view [40], those involved in managing invasive species have pointed out that benefits from such species are often overlooked or under-reported [39]. Our results, which provide the first information on the socio-economic impact of a parthenogenetic decapod described as 'the perfect invader' [22], provide a useful case study in how people make use of a new species in their environment. It also provides information useful for predicting the likely future spread of this invasive crayfish in a country with freshwater biodiversity of global value [41].

Madagascar is one of the poorest countries in the world, with 75% of the population living on less than $1.90 per day [42,43]. Poverty manifests itself in limited intake of dietary protein leading to stunting [44] which affects 1 in 2 Malagasy children [45]. In fact, many people rarely consume animal protein [35,36]. Given the lack of sufficient protein for human consumption; livestock are mostly fed rice bran, roots and tubers [46], which provides insufficient protein for livestock to put on weight effectively. New and cheap sources of animal protein can create a significant beneficial impact on humans. Positive perceptions of marbled crayfish (especially prevalent in the Analamanga region) are associated with harvesting, buying, selling or using marbled crayfish for animal feed. The fact that the crayfish can be collected from public lakes, rivers, canals and rice fields also means that it can be done cheaply. Live animals are sold at low prices (live marbled crayfish rank among the cheapest sources of animal protein available), which makes them affordable for poorer people. Leftover parts from processed marbled crayfish are fed to animals (mainly chicken and pigs); providing an additional source of income for those involved in the crayfish trade and providing an effective, cheap and protein-rich livestock feed.

In Antananarivo, the marbled crayfish has been common for at least 10 years [22]. The market has evolved and today the crayfish are mostly sold as peeled tail meat. This value-added product (similar to processed fish and pre-cut vegetables) increases the profit margin of crayfish. A different market exists in Ihorombe, where the animals appeared considerably later and most trade is for live crayfish in bulk. While marbled crayfish are seldom among respondent's

most preferred food, they are ranked similarly to shrimp and factory processed chicken. The low price, the ease of availability, and the development of new culinary recipes for marbled crayfish could be drivers of their increasing acceptance.

The most significant negative impacts reported were on rice farming and fishing. A related species of crayfish (*Procambarus clarkii*) is known to negatively affect irrigated rice production in the Iberian Peninsula, where it damages banks and irrigation canals by burrowing [47,48]. However, the extent to which rice production is impacted by marbled crayfish in Madagascar remains poorly understood and our survey results, associated qualitative information, provided some contradictory evidence. In Analamanga (where the crayfish have been present over the longest period of time), the largest proportion of respondents reported no impact, while some reported a positive impact, perhaps because of the impact of the crayfish on soil aeration. It is possible that reported negative impacts of crayfish stem from extensive media reporting of possible negative impacts on rice, rather than firsthand experience. Indeed, national daily newspapers such as Midi Madagasikara and L'Express de Madagascar, and local newspapers such as Tia Tanindrazana, started to describe the marbled crayfish as a threat to rice farming and as a national threat from 2007 [49–52]. The available evidence suggests negative impacts on fishing, but further research is needed to understand the impacts on fish and freshwater ecosystems.

The overall perception of marbled crayfish impacts differed among the participants in the three regions that were surveyed. Individuals in the Analamanga region perceived crayfish more favorably than those in the Mahatsiatra Ambony and Ihorombe regions. This may be due to differences in livelihoods of our samples in the three regions. Those interviewed in the Mahatsiatra Ambony and Ihorombe regions relied more on farming, particularly rice farming. Perceived negative impacts on rice farming are likely to influence the overall negative perception. Another possible explanation is that people in Analamanga (where marbled crayfish have been established the longest; since about 2005) have had greater exposure to the species, thus more time to adapt.

It is interesting that we noted such high acceptability of the crayfish as food, especially around Antananarivo, given the negative cultural associations which have developed since the species first appeared. The local name for the crayfish in Malagasy ("foza orana"; literally, crab-like crayfish) quickly took on the meaning of something which is poor in quality and very abundant. For example, the term has been applied to many things such as poor quality cell phones, and it is used as an insulting term for sex workers [53].

The geographical range of many invasive species is driven by socio-economic factors, such as trade, travel and the movement of goods [54,55]. Marbled crayfish were sold in markets in Madagascar soon after they first appeared. This raised the possibility that their potential economic importance could promote invasiveness [22]. Our work clearly confirms that the market demand for the crayfish is likely to encourage spread into new areas. Even in Ihorombe, where the overall perception of the crayfish is negative, the strong market demand and widespread distribution of live animals are likely to favor further spread.

Lastly, it is important to point out some limitations of our analysis. The sampling approach used, and the relatively small numbers of interviews, means that our sample is not representative of the population in each region. However, probabilistic sampling was not possible, given our limited resources and the lack of a suitable sampling frame. Our sample included several points in each region (see maps in S1 Data of S2 Fig) and the high response rate means that the responses are not likely to be biased. Furthermore, we controlled for differences in age and gender.

## Conclusion

The spread of the marbled crayfish in Madagascar over the last decade has been nothing short of dramatic. Given the practical challenges to eradicate this invasive species, a balanced

approach for considering the costs of benefits arising from the invasion is needed [39]. There are differing perceptions towards marbled crayfish; between regions and between people with different socio-economic characteristics. However, it is clear that the market for marbled crayfish, and their widespread consumption, are likely to result in further spread. It is difficult to imagine any inhabitable parts of Madagascar that will remain uninvaded over the next decade. What this will mean for Madagascar's freshwater biodiversity is unknown. This study has attempted to provide some insight into what impact the invasion has on local people's livelihoods. An important area of uncertainty remains the impact of marbled crayfish on rice agriculture. Given the importance of irrigated rice production to the economy of Madagascar, this is certainly worthy of further investigation. Our study demonstrated that marbled crayfish have become a widely consumed and cheap source of animal protein in local diets. Given the desperate need for protein in the diets of many Malagasy people, this can be considered a positive development.

## Supporting information

**S1 Data.**
(DOCX)

## Acknowledgments

We thank Maevatiana N. Ratsimbazafindranahaka and Luticia V. Raharimalala for their valuable support of our survey, and Amy Lewis for help with coding the survey in Open Data Kit. We also thank Soatiana Daniel and Laza S. Andriantsoa for their help with field trip management and anonymous reviewers whose suggestions improved the final manuscript.

## Author Contributions

**Conceptualization:** Ranja Andriantsoa, Julia P. G. Jones.

**Data curation:** Ranja Andriantsoa.

**Formal analysis:** Ranja Andriantsoa, Vlad Achimescu.

**Investigation:** Ranja Andriantsoa, Heriniaina Randrianarison, Miary Raselimanana, Manjary Andriatsitohaina, Jeanne Rasamy.

**Methodology:** Julia P. G. Jones, Vlad Achimescu.

**Supervision:** Ranja Andriantsoa, Julia P. G. Jones, Vlad Achimescu, Jeanne Rasamy, Frank Lyko.

**Visualization:** Ranja Andriantsoa.

**Writing – original draft:** Ranja Andriantsoa, Julia P. G. Jones, Frank Lyko.

**Writing – review & editing:** Julia P. G. Jones, Frank Lyko.

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
