## [Decision Letter · Decision Letter 0]

9 Jan 2020

PONE-D-19-30933

Socio-economic impacts of the marbled crayfish invasion in Madagascar

PLOS ONE

Dear Dr. Lyko,

Thank you for submitting your manuscript to PLOS ONE. After careful consideration, we feel that it has merit but does not fully meet PLOS ONE’s publication criteria as it currently stands. Therefore, we invite you to submit a revised version of the manuscript that addresses the points raised during the review process.

All comments offered by external reviewers and my own shall be fully addressed. The study and findings described in the manuscript can make a valuable contribution to the field, but to do so a revised version responding to our collective insights is deemed necessary. 

We would appreciate receiving your revised manuscript by Feb 23 2020 11:59PM. To enhance the reproducibility of your results, we recommend that if applicable you deposit your laboratory protocols in protocols.io, where a protocol can be assigned its own identifier (DOI) such that it can be cited independently in the future. For instructions see: http://journals.plos.org/plosone/s/submission-guidelines#loc-laboratory-protocols

We look forward to receiving your revised manuscript.

Kind regards,

Francisco X Aguilar

Academic Editor

PLOS ONE

Journal Requirements:

2. In your Methods section, please provide additional location information of the sampling sites, including geographic coordinates for the data set if available

Additional Editor Comments:

Three reviewers have offered valuable comments to the original version of the manuscript.

Authors should pay particular attention to the detailed review submitted by Reviewer 1. They will significantly improve the structure and readability of the manuscript.

In addition to the reviewers' observations I will ask the authors to:

- Offer a test-statistic that can validate the use of OLS regression regarding the normal distribution (or not) of their dependent variable. If normality it rejected the results of their regression will be highly questionable.

- The authors should also include p-values from t-tests regarding whether mean values and coefficients, respectively, are different from 0. These should be added to Figures 2 and 3.

- Details should be disclosed on their type of ordinal regression (e.g. ordinal logit/probit, tobit, other?). These model specifications all offer p-values for each coefficient. None are presented in the Appendix.

- Results from their main regression model should be reported within the main body of the paper. This is a major result that should not be placed within Appendices.

- Typo in S3 table (repeated) should be corrected.

Reviewers' comments:

Reviewer's Responses to Questions

**Comments to the Author**

1. Is the manuscript technically sound, and do the data support the conclusions?

Reviewer #1: Yes

Reviewer #2: Yes

Reviewer #3: Yes

2. Has the statistical analysis been performed appropriately and rigorously? 

Reviewer #1: I Don't Know

Reviewer #2: Yes

Reviewer #3: Yes

3. Have the authors made all data underlying the findings in their manuscript fully available?

Reviewer #1: Yes

Reviewer #2: Yes

Reviewer #3: Yes

4. Is the manuscript presented in an intelligible fashion and written in standard English?

Reviewer #1: Yes

Reviewer #2: Yes

Reviewer #3: Yes

5. Review Comments to the Author

Reviewer #1: This is an interesting study that explores mixed perceptions about the marbled crayfish invasion in Madagascar. I especially like the balanced approach to invasion ecology, instead of simply focusing on the negative aspects. I think it is publishable given proper attention to detail (mainly grammar). I have made numerous copy-edits to improve the quality of expression.

Reviewer #2: Overall, this is an interesting study, providing insights into the value of invasive species. While the study focuses mostly on a specific species in Madagascar, it has broad implications for our perception of invasive species.

Line 50-52: I'm not sure what this is or why is it here

Line 76: You should also provide the vernacular name of this species as you did with all other species in this paragraph

Line 81: where is native of?

Line 86-88: it would be better to have a map here to show the location of Antananarivo, the distribution range, and the 15 sites. Also, here, you mention 15 regions studied, it is not clear whether this refers to the present study or the study in reference #28. If it refers to this present study, it is more confusing because later on (line 121 for example), it seems as if you conducted your research in only 3 regions; but then you discuss other regions that were not mentioned in the method sections and then in the results you have 18 regions.

Line 81-96: It would be nice to know when was the marbled crayfish introduced in Madagascar, from where and for what reason it was initially introduced (if known). Was it intentional or accidental introduction? Also, has there been any actions toward its management and control given the concerns regarding its potential negative impacts on rice agriculture and fish populations?

Line 119-121: Figure 1A? Also, who did the survey in 2017 and where? As I mentioned above, the total number of regions in the present study is not clear.

Line 121-125: Please consider adding information on the following points to better explain your study design: What do you refer to as a "sampling station"? How far were the sampling stations from each other? What type of water body? What exactly did you sample? Why did you select these particular blocks of time? How many sampling efforts were done within each block of time and/or in total? What did you do to the captured animals? Was the sampling three days per station or three days per region? It would be nice to have a picture of "tandroho" in supplementary info since not everyone may be familiar with that.

Line 129: Why did you select only these sites for the survey details?

Line 136: it may be better to provide a short description of the Likert survey method.

Line 167-168: I would suggest moving this last sentence earlier (maybe near line 133).

Line 193-194: Add a map of the range

Line 195: what do you mean by "negative sites"?

Line 198: provide a mean and/or range of the density

Line 201-203: These sampling points seem to be more appropriate to be moved into the method section (line 129)

Line 229-230: it is not clear from the description of the survey the meaning of "negative/positive attitude"

Line 241: burrowing activities in marbled crayfish (maybe add that last phrase to make it clear that you're referring to the animal when taking about the burrowing)

Line 305: For consistency with the rest of the manuscript, maybe add also the equivalence of this amount in Euros

Line 366-369: This is more of a curiosity (and maybe my ignorance on the subject), but do you know if the leftover parts of the processed crayfish (I'm guessing the exoskeleton and head) actually provide additional protein (and in higher quantity) to the livestock?

Line 386: It would be nice to have an example of media reporting on negative impacts on rice (for example, a link to a local newspaper on that subject)

Line 416: I think invasion is more appropriate here than invasiveness

Figure S2 caption: Provide both (or either) vernacular and scientific names for the marbled crayfish and the various animal species presented here for consistency

Figure S3 caption: F: I think "traditionally reared/farmed chicken" is the appropriate wording

Reviewer #3: The authors should mention the social aspect of marbled crayfish in the Analamanga region where mostly only poor people rely on it as source of protein and many others have some "pejorative consideration" of this species,which means a kind of "seeing people eaten it as very poor" --- such consideration is not linked with its impact to the biodiversity but some type of "social and cultural" consideration locally known as "foza orana" : something bad - destroying etc. This mentality might have impact on limiting its consumption in the Analamanga region

6. PLOS authors have the option to publish the peer review history of their article (what does this mean?). If published, this will include your full peer review and any attached files.

Reviewer #1: No

Reviewer #2: No

Reviewer #3: Yes: Aristide ANDRIANARIMISA

---

## [Author Response · Author response to Decision Letter 0]

18 Feb 2020

Journal Requirements

>> The manuscript was edited in order to meet the PLOS ONE’s style requirements.

2. In your Methods section, please provide additional location information of the sampling sites, including geographic coordinates for the data set if available.

>> Geographical coordinates are now provided in the tables S1-3.

Additional Editor Comments

1. Authors should pay particular attention to the detailed review submitted by Reviewer 1. They will significantly improve the structure and readability of the manuscript.

>> Reviewer 1 provided edits on the manuscript itself. We have read all of these suggested edits and thought about them carefully (including considering why the reviewer may have felt our wording could be better expressed). We have made some edits in response. However many we have rejected as we preferred our own way of expressing these points (or we have edited and found a clearer way in our own words).

2. Offer a test-statistic that can validate the use of OLS regression regarding the normal distribution (or not) of their dependent variable. If normality it rejected the results of their regression will be highly questionable.

>> In order to validate the use of OLS, we assessed the normality of the residuals. While the dependent variable shows a bimodal distribution (Fig. S5A), the distribution of the residuals is approximately normal (Fig. S5B). We also used a Q-Q plot in order to check the assumption. The linearity of the points satisfies the assumption of normal distribution (Fig. S5C). Finally, we performed a Shapiro-Wilk normality test on the residuals which revealed a p-value=0.603 (>0.05); therefore the null-hypothesis of normal distribution cannot be rejected. These results have now been integrated into the manuscript.

3. The authors should also include p-values from t-tests regarding whether mean values and coefficients, respectively, are different from 0. These should be added to Figures 2 and 3.

>> Results in Figure 2 are descriptive. Therefore, no statistical test was performed on the data set. For Figure 3, the p-values are provided in Table 1. This is now clarified in the figure legend.

4. Details should be disclosed on their type of ordinal regression (e.g. ordinal logit/probit, tobit, other?). These model specifications all offer p-values for each coefficient. None are presented in the Appendix.

>> The ordinal regression we used is a proportional odds logistic regression (Agresti et al., 2002; Venables et al., 2002). This information was now added to the method section. The p-values for each coefficient for the ordinal regression are now provided in Table S4. 

5. Results from their main regression model should be reported within the main body of the paper. This is a major result that should not be placed within Appendices.

>> The regression table has been moved to the main body of the paper.

6. Typo in S3 table (repeated) should be corrected.

>> The typo in Table S3 (now Table S5) was corrected.

 

Reviewer 1

1. The manuscript is plagued with improper English and needs considerable attention to detail before it is acceptable for publication (i.e., redundancies, contradictions, verb tense, splitting verbs, misspellings, and run-on sentences). I made many edits throughout the paper (see PDF) for consideration in the next round. For example, why is it important to mention that a Likert scale was used 5 times? What is a substantial fraction (did you mean percent)? Why did you say, tilapia fish? The word ‘order’ is used twice in the same sentence; once as a noun and once as a verb.

>>The manuscript was carefully edited after reading the comments and suggestions. However we have not accepted the edits of our text when we felt these changed the style or voice in ways we didn’t feel comfortable with. 

2. Terminology, in some cases, was inconsistent with traditional usage in the social sciences. For example, it was a convenience sample, not haphazard or opportunistic. 

>> We have changed terminology in some places in response to these comments. 

3. Although the sampling period was identified (March and April 2019), it did not include information on weekday vs. weekend data collection.

>> Data was not exclusively collected on weekend of weekdays We have clarified this point. 

4. The perception of marbled crayfish was measured by 6 items (not aspects), yet a Cronbach’s alpha (reliability coefficient) was not provided.

>> While performing the data analysis, we did not create a sum score index and only considered the items separately; therefore, a Cronbach’s was not appropriate. For this reason, this information was not added to the manuscript.

5. The dependent variable (perception of impacts) was measured on a 5-point scale and referred to as ordinal data. Most social science research uses a 5-point scale and calls it interval data. Your choice of statistics hinges upon this decision. Please re-run your data using linear regression to see how the results compare with OR and OLS. Perhaps you can improve R2 even more than it is already.

>> In our analysis, we treated the 5 point-scale as interval data, and we checked for the residuals which are normally distributed. We used ordinal regression for sensitivity analysis. The results of the Ordinal Regression (OR) and Linear Regression (Ordinary Least Square) models are also comparable: coefficients that are positive in OLS are also positive in OR, same for the negative coefficients, and those not significantly different from 0 (p>0.05). This is described in Table S4.

6. I did not see a sample size of key informants, or a description on how you analyzed this data. I saw some direct quotes in the narrative, but it would be better if you had categorized the open-ended information so I could see it in a table. 

> We treated the whole data set together, as the same closed-ended questionnaire was used for all participants (see appendices which provides copies of the survey instruments). The only difference with the key informants is that they were asked additional open-ended questions. We used the quotes from these open-ended questions (52 participants out of 351) to complement our quantitative findings. We did not perform a separate analysis for the open-ended questions as the data set was small, particularly when split per region. We have added information on sample size to the results section.

 

Reviewer 2

We thank the reviewer for his/her attention to detail. The manuscript was carefully edited after reading the comments and suggestions. We provide three new tables to the supplementary information S1-3 in order to clarify the study locations and sampling details as suggested by this reviewer. Most comments were made on the manuscript. We respond to these below:

The reviewer didn’t like the use of the quote to open the manuscript.

>> This is commonly done in social sciences papers. Also, all authors really like the use of this special quote which captures the essence of the paper and places local people’s voices at the heart of the paper. The other reviewers and editor didn’t ask us to remove it and we would very much prefer to keep it. 

“What about people who did not like it? - you seem only get information for those who like the marbled crayfish but no mention of why others hate to eat it ?? so probably best to mention that the investigation was limited to some type of people in the region.” 

>> A detailed explanation of our sampling approach is provided in the manuscript. While it is not perfect, we have interviewed a wide range of people with different, region-specific perceptions.

Reviewer 3

The authors should mention the social aspect of marbled crayfish in the Analamanga region where mostly only poor people rely on it as source of protein and many others have some "pejorative consideration" of this species, which means a kind of "seeing people eaten it as very poor" --- such consideration is not linked with its impact to the biodiversity but some type of "social and cultural" consideration locally known as "foza orana" : something bad - destroying etc. This mentality might have impact on limiting its consumption in the Analamanga region

>> We thank the reviewer for this constructive comment. The negative associations that many people have for the marbled crayfish as “foza orana” is indeed interesting and we have added this to the text.

---

## [Decision Letter · Decision Letter 1]

9 Mar 2020

PONE-D-19-30933R1

Socio-economic impacts of the marbled crayfish invasion in Madagascar

PLOS ONE

Dear Dr. Lyko,

Thank you for submitting your manuscript to PLOS ONE. After careful consideration, we feel that it has merit but does not fully meet PLOS ONE’s publication criteria as it currently stands. Therefore, we invite you to submit a revised version of the manuscript that addresses the points raised during the review process.

The authors need to address all comments brought up by Reviewer 1 regarding methods, content and style of the manuscript. Authors shall also engage in a full proof-read of the manuscript for the correct use of English. 

We would appreciate receiving your revised manuscript by Apr 23 2020 11:59PM. To enhance the reproducibility of your results, we recommend that if applicable you deposit your laboratory protocols in protocols.io, where a protocol can be assigned its own identifier (DOI) such that it can be cited independently in the future. For instructions see: http://journals.plos.org/plosone/s/submission-guidelines#loc-laboratory-protocols

We look forward to receiving your revised manuscript.

Kind regards,

Francisco X Aguilar

Academic Editor

PLOS ONE

Additional Editor Comments:

Reviewer 1, who previously provided a very comprehensive review, deems some of his comments have not been fully addressed. I agree with that assessment and authors must address these observations for the manuscript to be published.

In addition:

- The title shall be updated to reflect the clear nature of the study. This research examined 'perceived' impacts. It did not assess impacts per se, but the perceptions of those afflicted by this invasive species. For instance, it could be updated to "Perceived socio-economic impacts of the marbled crayfish invasion in Madagascar"

- Reviewer 1 is also clear on the need for a complete proof-read of the manuscript and has provided additional comments as an Appendix. The authors shall proof-read the manuscript for the proper use of English. As a case in point, the last sentence in the Abstract illustrates several grammatical issues (e.g. data are plural) "While data on the biodiversity impacts of the marbled crayfish invasion in Madagascar is still completely lacking, this study provides insight into the socio-economic impacts of the dramatic spread of this unique invasive species in Madagascar."

Reviewers' comments:

Reviewer's Responses to Questions

**Comments to the Author**

1. If the authors have adequately addressed your comments raised in a previous round of review and you feel that this manuscript is now acceptable for publication, you may indicate that here to bypass the “Comments to the Author” section, enter your conflict of interest statement in the “Confidential to Editor” section, and submit your "Accept" recommendation.

Reviewer #1: (No Response)

2. Is the manuscript technically sound, and do the data support the conclusions?

Reviewer #1: Yes

3. Has the statistical analysis been performed appropriately and rigorously? 

Reviewer #1: No

4. Have the authors made all data underlying the findings in their manuscript fully available?

Reviewer #1: (No Response)

5. Is the manuscript presented in an intelligible fashion and written in standard English?

Reviewer #1: No

6. Review Comments to the Author

Reviewer #1: I like this study. It makes a valuable contribution to invasion ecology by introducing some benefits associated with the presence of invasive species. I look forward to seeing it published soon. Two issues remain.

Despite having "carefully edited" the manuscript, the authors still need to address a number of grammatical problems / issues before it can be published. I have addressed many of them (see the copy edits). Although some of my suggestions are stylistic (subjective), others are objective (not subject to what you feel comfortable with). For example, splitting the verb (have recently been observed) is improper English. It's a problem throughout your manuscript.

The second issue involves the dependent variable. Social science does not support the use of one "global" indicator for overall perception. You measured six items, and they should be summed to form an overall perception. That's the purpose of the scale, hence a Cronbach's alpha is a needed and appropriate statistic to include in your analysis.

7. PLOS authors have the option to publish the peer review history of their article (what does this mean?). If published, this will include your full peer review and any attached files.

Reviewer #1: No

---

## [Author Response · Author response to Decision Letter 1]

24 Mar 2020

Editor comments

1. Reviewer 1, who previously provided a very comprehensive review, deems some of his comments have not been fully addressed. 

>> We have now fully addressed the points that were raised by Reviewer 1.

2. The title shall be updated to reflect the clear nature of the study. This research examined 'perceived' impacts. It did not assess impacts per se, but the perceptions of those afflicted by this invasive species. For instance, it could be updated to "Perceived socio-economic impacts of the marbled crayfish invasion in Madagascar"

>> Done as suggested.

3. Reviewer 1 is also clear on the need for a complete proof-read of the manuscript and has provided additional comments as an Appendix. The authors shall proof-read the manuscript for the proper use of English. As a case in point, the last sentence in the Abstract illustrates several grammatical issues (e.g. data are plural) "While data on the biodiversity impacts of the marbled crayfish invasion in Madagascar is still completely lacking, this study provides insight into the socio-economic impacts of the dramatic spread of this unique invasive species in Madagascar."

>> We have accepted as many of the suggested edits as possible. However, we do somewhat object to the implication that our English was not of high quality and we are holding out on some suggested edits. Please also see our response to Reviewer 1 below and the cover letter by Julia Jones.

Reviewer 1

1. “Despite having "carefully edited" the manuscript, the authors still need to address a number of grammatical problems / issues before it can be published. I have addressed many of them (see the copy edits). Although some of my suggestions are stylistic (subjective), others are objective (not subject to what you feel comfortable with). For example, splitting the verb (have recently been observed) is improper English. It's a problem throughout your manuscript.”

>> The reviewer is understandably frustrated that (s)he spent a lot of time editing our manuscript and we did not accept the edits. After the first round of reviews, the author team decided not to accept many of the edits suggested by this reviewer because the 2nd author (a native speaker) did not agree with this reviewer’s stylistic edits and argued to retain many of our formulations. Perhaps we should have explained this decision more clearly in our response after the 1st round. This time, as the reviewer is insisting on the edits and the editor has supported them, we have accepted the majority of the edits (more than 130 out of the 144 edits that were made). However, we have rejected some edits that would have changed the meaning or the emphasis of the contents of our manuscript. For example, the reviewer wanted us to change ‘primary livelihood’ for ‘primary job’. This is not appropriate as in most cases local livelihood’s do not depend on a ‘job’ as usually understood. We have also rejected the edits to the first paragraph of the discussion.

2. “The second issue involves the dependent variable. Social science does not support the use of one "global" indicator for overall perception. You measured six items, and they should be summed to form an overall perception. That's the purpose of the scale, hence a Cronbach's alpha is a needed and appropriate statistic to include in your analysis.”

>> The reviewer is suggesting that the various items in our survey are measuring a latent variable which relates to a general attitude towards the crayfish. We have redone the analysis in the way the reviewer suggests. Reassuringly, the results confirmed our previous conclusions (as detailed below). The manuscript has been revised and updated correspondingly.

We changed the dependent variable from a one-item overall perception of crayfish impact to an index composed of 4 items: overall perception, impact on household economy, impact on food security, impact on health. We did not include the other impact variables (rice farming, animal feed and fishing), because they were not presented to the full sample, but only shown to respondents engaged in these activities. We wanted to keep the balanced sample that includes people with different livelihoods. Because there was missing data due to nonresponse, we imputed scale values for some cases, but only for the respondents with at least 2 of 4 valid (non-missing) item responses. We checked the 4-item scale reliability using Cronbach’s alpha, which showed a satisfying level of internal consistency. Finally, the index was standardized to a mean of 0 and standard deviation of 1 to simplify the interpretation of the regression coefficients.

 We then ran the linear regression (OLS) with the new dependent variable (crayfish impact index), and checked the residuals for violations of assumptions (there were none). We removed the ordinal regression sensitivity check, since it was not adequate anymore (the dependent variable measurement level can be treated as interva ldata). The results were very similar to the ones where we used the overall perception index.

---

## [Editor Report · Decision Letter 2]

1 Apr 2020

Perceived socio-economic impacts of the marbled crayfish invasion in Madagascar

PONE-D-19-30933R2

Dear Dr. Lyko,

We are pleased to inform you that your manuscript has been judged scientifically suitable for publication and will be formally accepted for publication once it complies with all outstanding technical requirements.

With best regards,

Francisco X Aguilar

Academic Editor

PLOS ONE

---

## [Editor Report · Acceptance letter]

2 Apr 2020

PONE-D-19-30933R2 

Perceived socio-economic impacts of the marbled crayfish invasion in Madagascar 

Dear Dr. Lyko:

I am pleased to inform you that your manuscript has been deemed suitable for publication in PLOS ONE. Congratulations! Your manuscript is now with our production department. 

With kind regards,

on behalf of

Dr. Francisco X Aguilar 

Academic Editor

PLOS ONE